# CAN GNNS LEARN HEURISTIC INFORMATION FOR LINK PREDICTION?

## ABSTRACT

Graph Neural Networks (GNNs) have shown superior performance in Link Prediction (LP). Especially, SEAL and its successors address the LP problem by classifying the subgraphs extracted specifically for candidate links, gaining state-of-the-art results. Nevertheless, we question whether these methods can effectively learn the information equivalent to link heuristics such as Common Neighbors, Katz index, etc. (we refer to such information as heuristic information in this work). We show that link heuristics and GNNs capture different information. Link heuristics usually collect pair-specific information by counting the involved neighbors or paths between two nodes in a candidate link, while GNNs learn node-wise representations through a neighborhood aggregation algorithm in which two nodes in the candidate link do not pay special attention to each other. Our further analysis shows that SEAL-type methods only use a GNN to model the pair-specific subgraphs and also cannot effectively capture heuristic information. To verify our analysis, a straightforward way is to compare the LP performance between existing methods and a model that learns heuristic information independently of the GNN learning. To this end, we present a simple yet light framework *ComHG*[1] by directly *Com*bining the embeddings of link *H*euristics and the representations produced by a *G*NN. Experiments on OGB LP benchmarks show that ComHG outperforms all top competitors by a large margin, empirically confirming our propositions. Our experimental study also indicates that the contributions of link heuristics and the GNN to LP are sensitive to the graph degree, where the former is powerful on sparse graphs while the latter becomes dominant on dense graphs.

## 1 INTRODUCTION

Link Prediction (LP), aiming at predicting the existence likelihood of a link between a pair of nodes in a graph, is a prominent task in graph-based data mining (Kumar et al., 2020). It has a wide range of beneficial applications, such as recommender systems (Wu et al., 2021), molecular interaction prediction (Huang et al., 2020), and knowledge graph completion (Li et al., 2022).

Throughout the history of LP research, a number of link heuristics have been defined, such as Common Neighbors (CN), Katz index (Katz, 1953), etc. A link heuristic usually describes a specific fact or hypothesis that gives the best interpretation to a statistical pattern in link observations (Martínez et al., 2016). The effectiveness of many link heuristics has been confirmed in various real-world LP applications (Liben-Nowell & Kleinberg, 2007; Zhou et al., 2009; Martínez et al., 2016).

Recently, graph representation learning has been proven powerful for LP (Perozzi et al., 2014; Zhang & Chen, 2018; Yun et al., 2021). Among the approaches in this domain, Graph Neural Networks (GNNs) have demonstrated stronger LP performance than others like node embedding methods based on positional encoding (Perozzi et al., 2014; Galkin et al., 2021). Modern prevalent GNNs like GCN (Kipf & Welling, 2017), GAT (Veličković et al., 2018), etc. follow a form of neighborhood information aggregation algorithm in which each node's representation is updated by aggregating the representations of this node and its neighbors. In this paper, we use the term GNNs to refer to such aggregation-based GNNs.

---

[1]Our code is available at `https://github.com/astroming/ComHG`

Figure 1: An illustration of the difference between link heuristics and aggregation-based GNNs.

In the literature, several LP-specific methods based on GNNs have been proposed, such as SEAL (Zhang & Chen, 2018), GraiL (Teru et al., 2020) and NBFNet (Zhu et al., 2021). In particular, SEAL and its follow-up works (Zhang & Chen, 2018; Li et al., 2020; Teru et al., 2020; Yin et al., 2022) predict the link likelihood between two nodes through classifying the subgraph extracted specifically for this target pair of nodes (we refer to such subgraph as *pair-specific subgraph*). SEAL-type methods also label every node in the pair-specific subgraph according to the relationship of the node to the target pair of nodes (Zhang et al., 2021). The pair-specific subgraphs together with the node labeling help SEAL-type methods learn better link representations than other methods, attaining state-of-the-art LP performance.

Despite all successes achieved by existing GNN-based LP methods, we are still curious about the question: *whether these methods can effectively learn the information equivalent to link heuristics (i.e., heuristic information) for LP?* Our analysis and experiments suggest a negative answer.

**Contributions.** In this work, we show that traditional link heuristics and GNNs capture different information. As illustrated in Figure 1, link heuristics are typically defined based on the number of involved nodes or paths between a pair of nodes. They are pair-specific. By comparison, a GNN updates the representation of a node by aggregating the representations of this node and its neighbors, where none of the neighbors is treated particularly. The learned nodes' representations are node-level. GNNs pay more attention to what information every node has, while link heuristics focus more on how many shared neighbors or paths are between a pair of nodes.

The difference between link heuristics and GNNs means that classical GNN-based LP methods simply combining the node-wise representations of two nodes in a candidate link into a link representation can hardly learn heuristic information. Moreover, we find that SEAL-type methods also cannot effectively capture heuristic information. Briefly, despite the help of the pair-specific subgraph and node labeling techniques, SEAL-type methods still use a GNN (e.g., DGCNN (Zhang et al., 2018) in SEAL (Zhang & Chen, 2018), R-GCN (Schlichtkrull et al., 2018) in GraiL (Teru et al., 2020)) to perform graph representation learning, where the GNN inherently lacks the model ability to learn heuristic information. Meanwhile, the labeling features of nodes are mixed in the neighborhood aggregation process of the GNN and the heuristic information embedded in the labeling features cannot be effectively kept in the learned node representations.

A simple way to verify our propositions is to study the LP performance of a model that separates the heuristic information learning and the GNN-based representation learning. Therefore, we present a light LP framework *ComHG* by *Com*bining link *H*euristics and the *G*NN. In ComHG, various link heuristics are encoded into trainable embeddings and combined with the representations produced by a GNN, followed by a predictor that takes the combinations as input to perform the final prediction. We conduct experiments on four OGB LP benchmark datasets (Hu et al., 2020). ComHG significantly outperforms all previous methods on all datasets. The strong results confirm that link heuristics and the GNN capture different yet effective information for LP, and suggest that combining both of them can boost LP performance. The results also empirically verify our analysis of the limitations of existing GNN-based LP methods in learning heuristic information. Furthermore, our experimental study shows that link heuristics could contribute more to LP performance on sparse graphs while GNN-based representation learning becomes dominant on dense graphs.

## 2 PRELIMINARIES

Without loss of generality, we demonstrate our work on homogeneous graphs. Let $\mathcal{G} = (\mathbb{V}, \mathbb{E})$ denote a graph $\mathcal{G}$ with $N$ nodes, where $\mathbb{V}$ is the set of nodes, $|\mathbb{V}| = N$, and $\mathbb{E}$ is the set of edges between the nodes in $\mathbb{V}$. A set of nodes connected directly to a node $v \in \mathbb{V}$ is the first-order neighborhood set of $v$ and is denoted as $\Gamma_v$. The degree of a node $v$ is defined as the number of edges connected to this node. The degree of a graph is defined as the average degree of all nodes.

**Problem formulation.** Link prediction is a pair-specific problem, aiming to estimate the likelihood of the existence of an unknown edge $\mathcal{E}_{vu} \notin \mathbb{E}$ between two nodes $v, u \in \mathbb{V}$. We denote the existence likelihood of $\mathcal{E}_{vu}$ as $\hat{y}_{vu}$ and refer to nodes $v, u$ as the *target pair of nodes*.

**Link heuristics.** Link heuristics mainly extract the structural information between a pair of nodes (Martínez et al., 2016). Shortest Path Distance (SPD) is a basic link heuristic that follows the underlying intuition that the distance between two related nodes should be short. Table 1 lists five most widely-used link heuristics. Specifically, Common Neighbors (CN) is defined as the size of the intersection between the first-order neighborhood sets of two nodes. Jaccard coefficient (Jaccard, 1901) normalizes the CN by the size of the union of the two nodes' neighborhood sets. AdamicAdar (Adamic & Adar, 2003) and Resource Allocation (RA) (Zhou et al., 2009) both suppress the contribution of nodes by penalizing each node in common neighbors with its degree. Katz index (Katz, 1953) counts all paths between

Table 1: Classical link heuristics for the node pair $(v, u)$.

| Heuristic | Definition |
|---|---|
| CN | $\lvert \Gamma_v \cap \Gamma_u \rvert$ |
| Jaccard | $\frac{\lvert \Gamma_v \cap \Gamma_u \rvert}{\lvert \Gamma_v \cup \Gamma_u \rvert}$ |
| AdamicAdar | $\sum_{z \in \Gamma_v \cap \Gamma_u} \frac{1}{\log \lvert \Gamma_z \rvert}$ |
| RA | $\sum_{z \in \Gamma_v \cap \Gamma_u} \frac{1}{\lvert \Gamma_z \rvert}$ |
| Katz | $\sum_{l=1}^{\infty} \beta^l \lvert \{ \mathrm{path}_{vu}^{(l)} \} \rvert$ |

a pair of nodes with weights, where $\lvert \{ \mathrm{path}_{vu}^{(l)} \} \rvert$ is the size of the set of all paths between node $v$ and $u$ with the length of $l$, and $\beta$ is a damping factor where $0 < \beta < 1$.

**Definition 1.** *The core function of most link heuristics is the set operation of two sets related to two nodes. We define the generalized function for such link heuristics as*

$$\mathrm{Heuristic}(v, u \mid v, u \in \mathbb{V}) = f\left(\mathrm{SETOP}(\mathbb{S}_v, \mathbb{S}_u)\right), \tag{1}$$

*where* $\mathrm{SETOP}(\cdot, \cdot)$ *is a set operation (e.g., intersection) applied on two sets.* $\mathbb{S}_v$ *and* $\mathbb{S}_u$ *are the sets related to node $v$ and $u$, respectively.* $f(\cdot)$ *is a function applied on the result of* $\mathrm{SETOP}(\cdot, \cdot)$.

Many commonly-used link heuristics can be expressed with Equation 1. For example, in CN, $\mathbb{S}_v$ is $\Gamma_v$, $\mathbb{S}_u$ is $\Gamma_u$, $\mathrm{SETOP}(\cdot, \cdot)$ is the intersection of $\mathbb{S}_v$ and $\mathbb{S}_u$, and $f(\cdot)$ obtains the cardinality of the set given by $\mathrm{SETOP}(\mathbb{S}_v, \mathbb{S}_u)$. In Katz index, $\mathbb{S}_v$ is the set of all paths that contain node $v$, $\mathbb{S}_u$ is the set of all paths that contain node $u$, and $\mathrm{SETOP}(\mathbb{S}_v, \mathbb{S}_u)$ outputs $\{ \mathrm{path}_{vu} \}$ that is the set of all paths between node $v$ and $u$. $f(\cdot)$ sums $\beta^l \lvert \{ \mathrm{path}_{vu}^{(l)} \} \rvert$ over different path length $l$, where $\{ \mathrm{path}_{vu}^{(l)} \} \subseteq \{ \mathrm{path}_{vu} \}$.

**Definition 2** (Heuristic information). *In this work, we say that the information captured by link heuristics is heuristic information.*

**Graph Neural Networks (GNNs).** Modern GNNs iteratively update the representation of each node in a graph by aggregating representations of its neighbors and its own (Corso et al., 2020). Formally, the representation of a node $v$ given by the $l$-th layer of a GNN is

$$\boldsymbol{h}_v^{(l)} = \mathrm{AGGREGATE}^{(l)}\left(\left\{ \boldsymbol{h}_{v'}^{(l-1)} \mid \forall v' \in \Gamma_v \cup \{v\} \right\}\right), \tag{2}$$

where $\mathbf{h}_v^{(0)}$ is initialized with the feature vector of node $v$, the function $\mathrm{AGGREGATE}(\cdot)$ is instantiated as a pooling operation over a set of node representations, such as MAX, MEAN, or attention-based SUM (Hamilton et al., 2017; Veličković et al., 2018). For simplicity, we omit the residual connections, activation functions, etc. In this paper, we use the term GNNs to refer to such aggregation-based GNNs unless otherwise stated.

**Remark 1.** *The representations learned by GNNs are node-wise.*

## 3 CAN EXISTING GNN-BASED LP METHODS EFFECTIVELY LEARN HEURISTIC INFORMATION?

In this section, we first show the difference between link heuristics and GNNs, and then present our analysis of the limitations of existing GNN-based LP methods in learning heuristic information.

**Proposition 1.** *The information captured by link heuristics is different from that embedded in the node representations learned by GNNs.*

*Proof.* Link heuristics are pair-specific. The $\text{Heuristic}(v, u \mid v, u \in \mathbb{V})$ in Equation 1 is specific for the node pair $(v, u)$, where the core function is a set operation of two sets $(\mathbb{S}_v, \mathbb{S}_u)$. By comparison, a GNN in Equation 2 learns a node-wise representation $\boldsymbol{h}_v^{(l)}$ based on a pooling operation over all elements in the set $\{\boldsymbol{h}_{v'}^{(l-1)} \mid v' \in \Gamma_v \cup \{v\}\}$, where the GNN does not pay special attention to any particular node in $\Gamma_v$. In other words, two nodes in a candidate link do not pay special attention to each other in the GNN learning. The information extracted through a set operation of two sets is fundamentally different from that obtained by a pooling operation over the elements in one set.

In addition, link heuristics mainly extract structural information between a pair of nodes by calculating the number of shared nodes or paths. They generally take no account of node features. In contrast, GNNs learn node-level representations that are initialized with node features, where the structural information of the graph is mainly used for generating the neighborhood set $\Gamma_v$. $\square$

According to Proposition 1, we have the following corollary.

**Corollary 1.** *GNNs inherently lack the model ability to learn heuristic information, i.e., the representation of each node learned by a GNN lacks heuristic information.*

### 3.1 CLASSICAL GNN-BASED LP METHODS

A direct way of applying a GNN to LP is to combine the learned node-wise representations of two nodes in a candidate link into a pair-wise link representation and then pass it into a predictor. We refer to such applications as *classical GNN-based LP methods* (Hamilton et al., 2017; Veličković et al., 2018; Wang et al., 2021). Formally, the link likelihood of node pair $(v, u)$ is

$$\hat{y}_{vu} = \text{PREDICTOR}\left(\text{COMBINE}\left(\boldsymbol{h}_v^{(L)}, \boldsymbol{h}_u^{(L)}\right)\right), \tag{3}$$

where $L$ is the index of the last GNN layer, $\text{COMBINE}(\cdot, \cdot)$ can be Hadamard production, concatenation, etc., and $\text{PREDICTOR}(\cdot)$ is a predictor like MLP.

**Proposition 2.** *Classical GNN-based LP methods lack the ability to capture heuristic information.*

*Proof.* The node-wise representations in Equation 3, i.e., $\boldsymbol{h}_v^{(L)}, \boldsymbol{h}_u^{(L)}$, are learned by a GNN. According to Proposition 1 and Corollary 1, both $\boldsymbol{h}_v^{(L)}$ and $\boldsymbol{h}_u^{(L)}$ lack heuristic information. Meanwhile, the function $\text{COMBINE}(\cdot, \cdot)$ in Equation 3 can hardly capture heuristic information like Equation 1, since this function does not involve any set operation on two sets related to node $v$ and node $u$. $\square$

Classical GNN-based LP methods in Proposition 2 are different from GNNs in Proposition 1. The former methods are downstream LP applications using the node representations learned by GNNs.

### 3.2 RETHINKING SEAL-TYPE METHODS

SEAL and its successors (Zhang & Chen, 2018; Li et al., 2020; Teru et al., 2020; Yin et al., 2022) address the LP problem by classifying the subgraphs that are extracted specifically for candidate links. We use the term *SEAL-type* methods to refer to these methods.

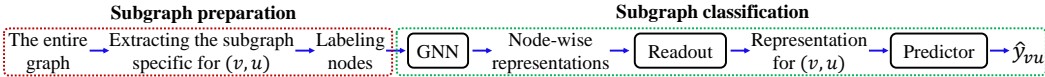

Figure 2: The algorithm flow chart of SEAL-type methods.

We briefly describe the algorithm flow of SEAL-type methods. As shown in Figure 2, given an entire graph $\mathcal{G}$ and a target node pair $(v, u)$, a $h$-hop enclosing subgraph $\mathcal{G}_{vu}^{(h)}$ with a set of nodes $\mathbb{V}_{vu}^{(h)} = \{v' \mid \text{Spd}(v', v) \leq h \text{ or } \text{Spd}(v', u) \leq h\}$ is extracted from $\mathcal{G}$, where $\text{Spd}(\cdot, \cdot)$ calculates the shortest path distance between two nodes. Then, for each node in $\mathcal{G}_{vu}^{(h)}$, a node labeling trick (Zhang et al., 2021) is used to assign a labeling vector to the node as its additional features according to its relationship to $(v, u)$. At the modeling stage, $\mathcal{G}_{vu}^{(h)}$ is fed into a GNN model, and the node-wise representations of the nodes in $\mathbb{V}_{vu}^{(h)}$ are learned. Following the last layer of the GNN, a readout function

READOUT($\cdot$) is employed over the learned node-wise representations of all nodes in $\mathbb{V}_{vu}^{(h)}$, and then a pair-specific representation for $\mathcal{G}_{vu}^{(h)}$ is produced. At last, a predictor PREDICTOR($\cdot$) takes this pair-specific representation as input to perform LP for the node pair $(v, u)$. Formally, the link likelihood of $(v, u)$ predicted by SEAL-type methods is

$$\hat{y}_{vu} = \text{PREDICTOR}\left(\text{READOUT}\left(\left\{\boldsymbol{h}_{v'}^{(L)} \mid \forall v' \in \mathbb{V}_{v,u}^{(h)}\right\}\right)\right). \tag{4}$$

The READOUT($\cdot$) (e.g., SortPooling in SEAL (Zhang & Chen, 2018)) is typically used for graph-level classification, aiming to deal with size differences among graphs.

SEAL-type methods usually use a node labeling trick to add labeling features to each node in the pair-specific subgraph. The labeling features of a node describe the relationship of the node to the candidate link. Figure 3 shows an example of this, where the labeling features of a node are the shortest path distances from the node to the target pair of nodes. Zhang et al. (2021) show that such labeling features can help a GNN learn better representations for LP.

Despite the benefits brought by the pair-specific sub-graph and node labeling techniques, our following analysis shows that SEAL-type methods still suffer from issues in learning heuristic information.

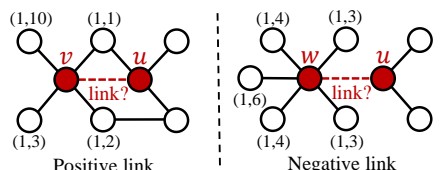

**Proposition 3.** *The labeling features of nodes contain heuristic information (e.g., SPD). However, the features of each node would be mixed with the features of other nodes in the neighborhood aggregation process of a GNN and thus the heuristic information in the labeling features cannot be effectively kept in the learned node representations.*

*Proof.* We use an example to illustrate this. As shown in Figure 3, the labeling features of a node in a pair-specific subgraph are SPDs from this node to the target pair of nodes. If the pooling method in a GNN is MEAN, then the aggregation of the labeling features of the neighbors of node $v$ would be equal to that of node $w$, i.e., $\text{MEAN}(\{10, 3, 2, 1\}) =$

Figure 3: Node labeling in SEAL-type methods. The left is a pair-specific sub-graph for a positive link sample and the right is a negative one. The labeling features are based on the SPDs from every node (here only show the first-order neighbors of node $v$ or $w$) to the target pair of nodes. For example, on the left, the node with the labeling features $(1, 10)$ indicates that the SPD from this node to node $v$ and $u$ is 1 and 10, respectively.

$\text{MEAN}(\{4, 6, 4, 3, 3\})$. This means that the labeling features of the neighbors of a node are mixed in the neighborhood aggregation process, and the distinct heuristic information in these labeling features is not effectively kept in the aggregated result. In other words, the aggregated results for node $v$ and $w$ in the positive and negative link samples become indistinguishable. Note that in practice, a GNN layer contains a series of complicated operations such as set-based pooling, linear and non-linear transformations, dropout, residual connection, etc. The heuristic information in the labeling features could be partially kept in the learned node representations. $\square$

Zhang et al. (2021) point out that the labeling features of nodes can help the GNN learn the heuristic information related to common neighbors. Their explanation is as follows. If node $v$ and $u$ are labeled, in the first iteration of the neighborhood aggregation in a GNN, only the common neighbors between node $v$ and $u$ will receive the labeling messages from both $v$ and $u$; then in the second iteration, these common neighbors will pass such messages back to both $v$ and $u$, which can encode the number of common neighbors into the representations of node $v$ and $u$. However, we question this statement. In the second iteration in the explanation above, in addition to the common neighbors between $v$ and $u$, the non-common neighbors of node $v$ also pass their messages back to $v$. The messages from all neighbors of $v$ are then aggregated through a set-based pooling (e.g., MEAN) as shown in Equation 2. Such set-based aggregated result for node $v$ inherently cannot retain the information related to the number of common neighbors. The same goes for node $u$. Therefore, we have the following proposition.

**Proposition 4.** *SEAL-type methods cannot guarantee that a GNN can use the node labeling features to effectively encode the neighborhood-based heuristic information into the node representations.*

Propositions 3 and 4 show that SEAL-type methods cannot effectively learn heuristic information embedded in the labeling features of nodes through GNNs. The main reason is that the GNNs

used in these methods are based on the neighborhood information aggregation algorithm (e.g., DGCNN (Zhang et al., 2018) in SEAL (Zhang & Chen, 2018), R-GCN (Schlichtkrull et al., 2018) in GraiL (Teru et al., 2020)). The distinguishing labeling features are mixed and become indistinguishable after the iterations of set-based neighborhood aggregation.

**Proposition 5.** *SEAL-type methods cannot effectively learn heuristic information for LP.*

*Proof.* We prove this by decomposing the entire algorithm of SEAL-type methods. As shown in Figure 2, the pair-specific subgraph extraction and the node labeling are the operations at the data preparation stage. Nevertheless, at the modeling stage, SEAL-type methods still use a GNN (e.g., DGCNN in SEAL (Zhang & Chen, 2018)) to learn node-wise representations (i.e., $\mathbf{h}_{v'}^{(L)}$ in Equation 4). Therefore, Proposition 1 and Corollary 1 still hold, and $\mathbf{h}_{v'}^{(L)}$ in Equation 4 lacks heuristic information. Furthermore, $\mathrm{READOUT}(\cdot)$ in Equation 4 can hardly capture heuristic information. $\mathrm{READOUT}(\cdot)$ is a pooling operation over all elements in one set (e.g., SortPooling in SEAL (Zhang & Chen, 2018)), which is different from the set operation of two sets used in link heuristics in Equation 1. Besides, Propositions 3 and 4 show that the heuristic information in the labeling features of nodes cannot be effectively kept in the learned node representations. $\square$

### 3.3 LIMITATIONS OF OTHER GNN-BASED LP METHODS

Most recently, Zhu et al. (2021) generalize several traditional path-based link heuristics into a path formulation and propose a novel neural network NBFNet to approximate the path formulation. Different from common GNNs that propagate and aggregate representations of nodes, NBFNet mainly utilizes the representations of edges between two nodes in a candidate link. NBFNet is a beneficial attempt to learn pair-wise heuristic information through representation learning. However, it suffers from some issues. First, NBFNet focuses on path-based heuristics like Katz index (Katz, 1953), while neglecting the neighborhood-based heuristics such as CN, RA (Zhou et al., 2009), etc. Second, the algorithm of NBFNet lacks the ability to utilize node-wise information such as node attributes and node embeddings. Besides, Yun et al. (2021) present Neo-GNN that weighted aggregates the LP score obtained by neighborhood-based heuristics and the score predicted by a GNN. However, Neo-GNN does not use path-based heuristics.

## 4 LEARNING HEURISTIC INFORMATION INDEPENDENTLY OF THE GNN

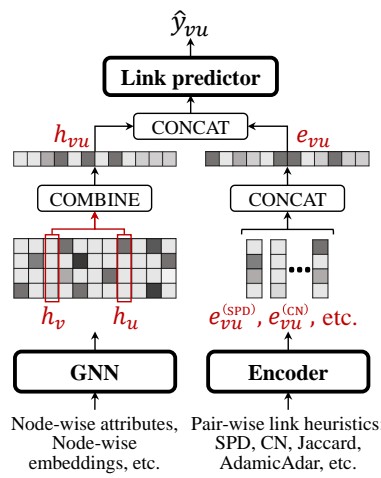

Figure 4: An overview of *ComHG*.

Propositions 1-5 show that traditional link heuristics and GNNs capture different information, and existing GNN-based LP methods fail to effectively learn heuristic information. We can validate these propositions by comparing the LP performance between existing methods and a model that learns heuristic information independently of the GNN learning. To this end, we present a simple yet light LP framework *ComHG* by *Com*bining link *H*euristics and the *G*NN.

### 4.1 *ComHG*: A SIMPLE YET LIGHT LP FRAMEWORK

As shown in Figure 4, ComHG mainly consists of two separate modules and a predictor. The GNN module produces a pair-level link representation by combining the node-level representations of the target pair of nodes. The heuristic module encodes various link heuristics into trainable embeddings and concatenates these embeddings. Then the link representation and the heuristic embedding are concatenated and passed into a predictor for the final prediction.

We emphasize that this work focuses on studying the limitations of existing GNN-based LP methods in learning heuristic information. The straightforward ComHG mainly serves as an examiner to validate our analysis. If Propositions 1-5 hold, ComHG would be expected to achieve comparable or superior performance to all existing LP methods.

**GNN module.** We use a GNN to perform graph representation learning. The GNN can be one of the existing GNNs such as GCN (Kipf & Welling, 2017), GAT (Veličković et al., 2018), or even MLP. We use node attributes, node embeddings, etc. as the initial node features. The node embeddings are trainable at the training stage. The GNN outputs node-wise representations for all nodes. We then combine the representations of each target pair of nodes into a pair-specific link representation. Formally, let $L$ be the index of the last GNN layer. The link representation for node pair $(v, u)$ is

$$\boldsymbol{h}_{vu} = \text{COMBINE}\left(\boldsymbol{h}_v^{(L)}, \boldsymbol{h}_u^{(L)}\right), \tag{5}$$

where $\text{COMBINE}(\cdot, \cdot)$ can be Hadamard production, concatenation, etc.

**Heuristic embedding.** Independently of the GNN module, ComHG encodes various link heuristics into trainable embeddings and then concatenates these embeddings. Let $e_{vu}$ be the concatenation of these embeddings for node pair $(v, u)$. We have

$$\boldsymbol{e}_{vu} = \text{CONCAT}(\boldsymbol{e}_{vu}^{(\text{SPD})}, \boldsymbol{e}_{vu}^{(\text{CN})}, \cdots, \boldsymbol{e}_{vu}^{(\text{RA})}), \tag{6}$$

where $\boldsymbol{e}_{vu}^{(\text{SPD})}, \boldsymbol{e}_{vu}^{(\text{CN})}, \boldsymbol{e}_{vu}^{(\text{RA})}$ is the embedding of SPD, CN, and RA, respectively. These embeddings are trained along with the entire framework. Note that ComHG can consider various link heuristics, which is beneficial to tackle the issue that some link heuristics may only be effective on some particular graphs (Kovács et al., 2019).

**Link predictor.** In ComHG, we concatenate the link representation $\boldsymbol{h}_{vu}$ and the heuristic embedding $\boldsymbol{e}_{vu}$ and pass the concatenation into a predictor. Formally, the link likelihood of $(v, u)$ is

$$\hat{y}_{vu} = \text{PREDICTOR}\left(\text{CONCAT}\left(\boldsymbol{h}_{vu}, \boldsymbol{e}_{vu}\right)\right). \tag{7}$$

We train ComHG using the loss based on negative sampling (Mikolov et al., 2013; Hamilton et al., 2017), i.e., $L = -\log(\hat{y}_{vu}) - \sum_{i=1}^{n} \frac{1}{n} \log(1 - \hat{y}_{v_i' u_i'})$, where the node pair $(v, u)$ is a positive link sample, $(v_i', u_i')$ is a negative sample, $n$ is the number of negative samples for each positive link.

**Computational complexity.** ComHG is computationally light. For the GNN module, ComHG can take advantage of existing light GNNs (Wu et al., 2019; Chiang et al., 2019). For link heuristics, most heuristics can be computed by matrix addition and multiplication on the adjacency matrix of a graph. The complexity of the dense matrix addition and multiplication is $O(N^2)$ and $O(N^3)$, respectively. In the sparse setting, these complexities can be reduced to $O(|\mathcal{E}|)$ (Gao et al., 2020), where $|\mathcal{E}|$ is the number of edges in a graph. The generation of link heuristics in ComHG is light compared to SEAL-type methods (Zhang & Chen, 2018; Teru et al., 2020). In SEAL-type methods, the complexity of subgraph generation can be approximated as $O(D^h|\mathcal{E}|)$, where $D$ is the degree of the entire graph and $h$ is the highest hop of the subgraph. We can see that the complexity of SEAL-type methods increases exponentially as the hop of the pair-specific subgraph increases.

In the following subsections, we present our experimental study.

## 4.2 EXPERIMENTAL SETUP

**Datasets.** We conduct experiments on four datasets: *ogbl-ddi, ogbl-collab, ogbl-ppa*, and *ogbl-citation2* from Open Graph Benchmark (OGB) (Hu et al., 2020). We **DO NOT** use previous widely-used datasets such as *Cora, Citeseer, Pubmed*, etc. because these datasets usually suffer from a series of issues such as unrealistic and arbitrary data splits, small scale, and data leakage. Especially, Shchur et al. (2018) show that different data splits on such datasets lead to different results in performance comparison among modern graph methods. By contrast, the OGB datasets are collected specifically for the LP task, which cover several realistic applications, span diverse scales (4K - 3M nodes), and come with standard evaluation procedures (Hu et al., 2020).

**Comparison methods.** We compare ComHG with several mainstream types of LP methods, including 1) heuristic methods: CN, AdamicAdar (Adamic & Adar, 2003), Katz (Katz, 1953); 2) graph representation learning methods (Non-GNN): Matrix Factorization (MF) (Menon & Elkan, 2011), DeepWalk (Perozzi et al., 2014), NodePiece (Galkin et al., 2021); 3) GNN methods: GCN (Kipf & Welling, 2017), GraphSAGE (Hamilton et al., 2017), GAT (Veličković et al., 2018), JKNet (Xu et al., 2018), LRGA+GCN (Puny et al., 2020), PLNLP (Wang et al., 2021), NBFNet (Zhu et al., 2021) and Neo-GNN (Yun et al., 2021); 4) SEAL-type methods: SEAL (Zhang & Chen, 2018),

Table 2: Results on test sets of OGB LP datasets. Higher is better. The best in each column is in bold. The $^\dagger$, $^\ddagger$, and $^\S$ indicate the top 1, 2, and 3 results given by previous methods.

| | Method | ogbl-ddi
Hits@20 (%) | ogbl-collab
Hits@50 (%) | ogbl-ppa
Hits@100 (%) | ogbl-citation2
MRR (%) |
|---|---|---|---|---|---|
| Non-GNN | CN | $17.73 \pm 0.00$ | $52.24 \pm 0.00$ | $27.60 \pm 0.00$ | $51.47 \pm 0.00$ |
| | AdamicAdar | $18.61 \pm 0.00$ | $54.09 \pm 0.00$ | $32.45 \pm 0.00$ | $51.89 \pm 0.00$ |
| | Katz | $15.32 \pm 0.00$ | $53.77 \pm 0.00$ | $26.28 \pm 0.00$ | $46.73 \pm 0.00$ |
| | MF | $13.68 \pm 4.75$ | $38.86 \pm 0.29$ | $32.29 \pm 0.94$ | $51.86 \pm 4.43$ |
| | DeepWalk | $26.42 \pm 6.10$ | $50.37 \pm 0.34$ | $23.02 \pm 1.63$ | $61.05 \pm 2.33$ |
| | NodePiece | $24.15 \pm 3.04$ | $47.88 \pm 0.41$ | $22.85 \pm 0.94$ | $61.52 \pm 1.59$ |
| GNN | GCN | $37.07 \pm 5.07$ | $44.75 \pm 1.07$ | $18.67 \pm 1.32$ | $84.74 \pm 0.21$ |
| | GraphSAGE | $53.90 \pm 4.74$ | $48.10 \pm 0.81$ | $16.55 \pm 2.40$ | $82.60 \pm 0.36$ |
| | GAT | $55.73 \pm 4.36$ | $42.51 \pm 2.92$ | $23.82 \pm 1.12$ | $79.97 \pm 1.13$ |
| | JKNet | $57.98 \pm 6.88$ | $48.84 \pm 0.83$ | $22.41 \pm 2.35$ | $84.28 \pm 1.38$ |
| | LRGA+GCN | $62.30 \pm 9.12^\S$ | $52.21 \pm 0.72$ | $26.12 \pm 2.35$ | $66.49 \pm 1.59$ |
| | PLNLP | $90.88 \pm 3.13^\dagger$ | $52.92 \pm 0.98$ | $32.38 \pm 2.58$ | $84.92 \pm 0.29$ |
| | NBFNet | $16.14 \pm 3.72$ | $51.05 \pm 0.38$ | $29.64 \pm 1.03$ | $51.22 \pm 6.17$ |
| | Neo-GNN | $63.57 \pm 3.52^\ddagger$ | $57.52 \pm 0.37^\dagger$ | $49.13 \pm 0.60^\ddagger$ | $87.26 \pm 0.84^\S$ |
| SEAL-type | SEAL | $30.56 \pm 3.86$ | $54.71 \pm 0.49^\ddagger$ | $48.80 \pm 3.16^\S$ | $87.67 \pm 0.32^\ddagger$ |
| | GraiL | $31.76 \pm 4.24$ | $54.19 \pm 0.52$ | $47.25 \pm 2.84$ | $86.59 \pm 0.58$ |
| | DEGNN | $26.63 \pm 6.82$ | $53.74 \pm 0.35$ | $36.48 \pm 3.78$ | $60.30 \pm 0.61$ |
| | SUREL | $32.31 \pm 4.15$ | $54.37 \pm 0.46^\S$ | $53.23 \pm 1.03^\dagger$ | $88.83 \pm 0.18^\dagger$ |
| Ours | ComHG(MLP) | $76.83 \pm 6.11$ | $54.75 \pm 2.52$ | $35.49 \pm 7.22$ | $85.61 \pm 1.83$ |
| | ComHG(GCN) | $91.38 \pm 3.08$ | $\mathbf{58.11 \pm 1.11}$ | $\mathbf{61.24 \pm 5.62}$ | $\mathbf{88.92 \pm 0.11}$ |
| | ComHG(GAT) | $\mathbf{92.15 \pm 3.62}$ | $58.03 \pm 1.21$ | $59.85 \pm 5.09$ | $88.23 \pm 0.38$ |

GraiL (Teru et al., 2020), DEGNN (Li et al., 2020), SUREL (Yin et al., 2022). OGB officially hosts a leaderboard[2]. We present the results of these methods as reported on this leaderboard or the original papers. More experimental settings are provided in the Appendix.

### 4.3 MAIN RESULTS.

The main experimental results are presented in Table 2. ComHG outperforms all top competitors and achieves state-of-the-art results on all datasets, demonstrating the effectiveness of the LP approach of learning heuristic information independently of the GNN learning.

As shown in Table 2, GNN-based LP methods usually perform better than non-GNN methods including heuristic methods and positional encoding-based embedding methods like DeepWalk and NodePiece. By comparison, ComHG outperforms all these methods, which could empirically support our Propositions 1, 2 and confirm that GNN-based representation learning and human-defined link heuristics capture different yet effective information for LP.

SEAL-type methods show stronger performance than other previous methods on three datasets, demonstrating their success. However, our Propositions 3, 4 and 5 indicate that such methods cannot effectively learn heuristic information. The results on ogbl-collab could empirically support these propositions. As shown on ogbl-collab in Table 2, heuristic methods including CN, AdamicAdar, and Katz significantly outperform most representation learning methods, indicating that heuristic information is critical for LP on this dataset. Meanwhile, SEAL-type methods are only slightly better ($0.62\%$ in SEAL, $0.1\%$ in GraiL, and $0.28\%$ in SUREL) or even worse ($-0.35\%$ in DEGNN) than the best heuristic AdamicAdar. By comparison, ComHG surpasses the AdamicAdar by $4\%$. The most likely explanation for this is that SEAL-type methods could not effectively learn the information equivalent to the link heuristics used in ComHG, considering that ComHG utilizes various link heuristics including SPD, CN, AdamicAdar, etc. Moreover, ComHG significantly exceeds the best SEAL-type methods by $60\%$ and $8\%$ on ogbl-ddi and ogbl-ppa, respectively. All these results demonstrate the superiority of the simple ComHG. The superiority is even more apparent when considering that ComHG is computationally lighter than SEAL-type methods.

---

[2]https://ogb.stanford.edu/docs/leader_linkprop/

We also evaluate three variants of ComHG where the GNN module uses MLP, GCN (Kipf & Welling, 2017) or GAT (Veličković et al., 2018). Table 2 shows that ComHG(MLP) performs worst across all datasets among three variants, which emphasizes the importance of neighborhood information aggregation of the GNN, considering the fact that MLP updates each node's representation independently of other nodes. We note that ComHG(GAT) performs worse than ComHG(GCN) on three datasets. This is probably because the graphs in the three datasets are large and the dimension of the representations used to compute the attention coefficient in GAT is too small, which limits the expressive power of GAT (a large dimension will dramatically increase the memory usage).

### 4.4 EFFECTS OF LINK HEURISTICS AND THE GNN.

During our experiments, we observe an interesting phenomenon in ComHG that the contributions of link heuristics and the GNN to the LP performance vary dramatically according to the graph degree. This phenomenon can also be seen in Table 2. For example, on ogbl-ddi with the graph degree of 500, the GNN-based methods such as GraphSAGE (Hamilton et al., 2017), GAT (Veličković et al., 2018), etc. significantly outperform the heuristic methods like CN and AdamicAdar. In contrast, on ogbl-collab with the graph degree of only 8, the simple heuristic CN and AdamicAdar methods perform even better than most representation learning methods.

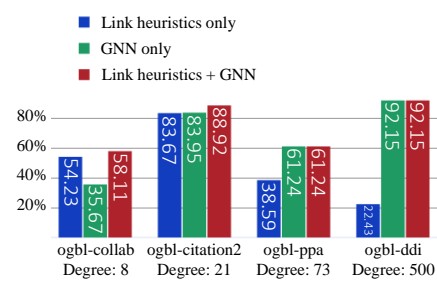

To further investigate this phenomenon, we study the effects of link heuristics and the GNN in ComHG. Figure 5 shows the results of ComHG using only link heuristics, only the GNN, and both of them. We can see that ComHG only using link heuristics performs better on the sparse graph (i.e., ogbl-collab) but worse on the dense graphs than ComHG only using the GNN. This trend is consistent with the results observed in Table 2. Moreover, we find that ComHG using both link heuristics and the GNN yields significant performance improvement on sparse graphs (i.e., ogbl-collab and ogbl-citation2), while on the dense ogbl-ppa and ogbl-ddi, ComHG only using the GNN can achieve the best LP results.

Figure 5: Performance of ComHG under different settings.

We explain these results as follows. For link heuristics, most link heuristics like CN, SPD, etc. are effective on sparse graphs but would become indistinguishable between positive and negative candidate links on dense graphs. For GNN-based representation learning, the representation of a node in a dense graph would be better learned than that in a sparse graph since more neighboring nodes would attend the neighborhood aggregation in the dense graph. For example, in a spare graph, a node $v$ has only one neighbor $w$. The learning of the representation of node $v$ in a GNN would only rely on the neighbor $w$. Then the learned representation of node $v$ would lack the ability to identify the relationships of $v$ to most of the other nodes in this sparse graph.

## 5 CONCLUSION

In this work, we showed that aggregation-based GNNs inherently lack the model ability to learn the information equivalent to traditional link heuristics. We provided an in-depth analysis of why existing GNN-based LP methods cannot effectively learn heuristic information. With the purpose of validating our analysis, we introduced a light LP framework ComHG that learns heuristic information independently of the GNN learning. The promising results achieved by ComHG empirically confirm our propositions and demonstrate the importance of heuristic information for LP.

**Limitations and future work.** We proved our propositions by analyzing the learning mechanism, algorithm architecture, and even an example. The proofs are not presented in a form of rigorous mathematical deduction. This is mainly because the node representations learned by GNNs are not easy to interpret. In addition, ComHG is developed mainly for validating our analysis. It simply combines the heuristic embedding and link representation separately produced by two modules. We believe that the power of integrating heuristic information with the GNN is worth exploring in the future, especially for multiple-node tasks.

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

# A    RELATED WORK

In this section, we briefly review several mainstream types of Link Prediction (LP) methods. Most methods used in performance comparison in our experiments are also described here.

**Heuristic methods.** Traditional link heuristics are usually defined based on the number of the neighbors or paths between a pair of nodes (Martínez et al., 2016). Most heuristic LP methods are developed for specific applications and their effectiveness has been confirmed in real-world tasks (Liben-Nowell & Kleinberg, 2007; Martínez et al., 2016; Kovács et al., 2019). In general, heuristic methods are interpretable (i.e., a link heuristic describes a specific fact or hypothesis to interpret a statistical pattern in link observations) and inductive (i.e., most link heuristics are independent of any particular node, and thus can naturally generalize to unseen nodes in the inductive LP (Zhu et al., 2021)). However, heuristic LP methods suffer from several issues. First, since many link heuristics are designed for specific graph applications, they may perform well only on a particular type of graphs (Kovács et al., 2019). Second, their model capacity is limited, considering that they mainly focus on the topological properties between target pairs of nodes but rarely leverage node features (Zhang & Chen, 2018).

**Node embedding methods based on positional encoding.** Node embedding methods are one class of graph representation learning methods. Such methods encode positional relationships between nodes into node embeddings. The similarity of nodes in the embedding space reflects the semantic similarity of nodes in the graph. These methods can partially encode the link heuristic information into node embeddings. Two nodes would have more similar embeddings if these two nodes are close to each other (path-based information) and have many common neighbors (neighborhood-based information) than the two nodes that are far apart. Following the word embedding method (Mikolov et al., 2013), the models such as Deepwalk (Perozzi et al., 2014), Node2vec (Grover & Leskovec, 2016), etc. learn node embeddings by treating the nodes as words and treating the sequences of nodes generated based on links as sentences. Most recently, inspired by subword tokenization used in NLP (Sennrich et al., 2016), NodePiece (Galkin et al., 2021) explores parameter-efficient node embedding techniques and presents an anchor-based method to learn node embeddings. However, all these methods typically train embeddings in an unsupervised learning manner. Solely using such node embeddings for LP empirically perform poorly compared to other representation learning methods (e.g., GNNs) and sometimes even worse than heuristic methods (Wang et al., 2021).

**Graph neural networks.** Recently, GNNs have demonstrated effectiveness in diverse real-world graph-based applications (Liang et al., 2021; Huang et al., 2020; Ying et al., 2021). A number of GNN models have been proposed. GCN  (Kipf & Welling, 2017) is a graph convolutional network that learns node representations by summing the normalized representations from the first-order neighbors. GraphSAGE (Hamilton et al., 2017) samples and aggregates representations from local neighborhoods with pooling methods like MEAN, MAX, LSTM, etc. GAT (Veličković et al., 2018) introduces an attention-based GNN architecture. LRGA (Puny et al., 2020) incorporates a Low-Rank global attention module to GNNs for improving their generalization power. These GNNs have demonstrated powerful LP performance by simply combining the node-level representations of two nodes in a candidate link into a link representation. Wang et al. (2021) present a framework PLNLP that jointly uses the node-wise representations learned by a GNN, node embeddings obtained by positional encoding methods, and even the link representations produced by SEAL-type methods.

**SEAL-type methods.** SEAL-type methods have shown superior performance among existing LP approaches (Hu et al., 2020; Teru et al., 2020). SEAL (Zhang & Chen, 2018) extracts a local enclosing subgraph for each candidate link and uses a GNN (Zhang et al., 2018) to classify these subgraphs for LP. GraiL (Teru et al., 2020) is developed for inductive knowledge graph reasoning. It is a method similar to SEAL but it replaces SortPooling readout with MEAN-pooling. DEGNN (Li et al., 2020) proposes a distance encoding GNN that comes with both theoretical guarantees and empirical efficiency. For LP, DEGNN uses a SEAL-type strategy but a different labeling technique. SUREL (Yin et al., 2022) proposes an algorithmic technique to improve the computational efficiency of subgraph generation in SEAL-type methods, where graph structures are decoupled into sets of walks that are used to generate pair-specific subgraphs.

These LP methods have achieved great success. However, our analytical and experimental results show that these existing methods lack the model ability to effectively capture heuristic information.

## B  MORE ON THE EXPERIMENTS

### B.1  IMPLEMENTATION DETAILS

We implement ComHG using PyTorch and PyTorch Geometric (Fey & Lenssen, 2019). The model is trained with Adam optimizer (Kingma & Ba, 2014). The learning rate is decayed following the ExponentialLR method. We conduct all experiments for ogbl-ddi and ogbl-collab on a Linux machine with 14-core CPU, 192G RAM, and NVIDIA Quadro P6000 (24G), and for ogbl-ppa and ogbl-citation2 on a machine with 32-cores CPU, 512G RAM and NVIDIA A100 (40G). Table 3 lists the configurations of ComHG for the best performance. We provide our code for reproducing the results at *https://github.com/astroming/ComHG*.

Table 3: Configurations of ComHG for the best performances.

|  | **ogbl-ddi** | **ogbl-collab** | **ogbl-ppa** | **ogbl-citation2** |
|---|---|---|---|---|
| GNN module | GAT | GCN | GCN | GCN |
| GNN layers | 2 | 2 | 2 | 2 |
| predictor | MLP | MLP | MLP | MLP |
| predictor layer | 4 | 5 | 3 | 4 |
| heuristics | - | SPD,CN,AA | - | SPD,AA |
| heuristic embedding dim | - | 32 | - | 32 |
| node embedding dim | 512 | - | 256 | - |
| lr | 0.003 | 0.002 | 0.001 | 0.001 |
| lr decay gamma | 0.995 | 0.997 | 0.99 | 0.995 |
| dropout rate | 0.3 | 0.3 | 0.3 | 0.25 |
| gradient clip norm | 5 | 10 | 5 | 10 |
| batch size | 100000 | 70000 | 100000 | 15000 |

### B.2  DATASETS

In this work, we do not compare the LP performance among methods on previous small graph datasets such as Cora, Citeseer, Pubmed (Zhu et al., 2021). This is mainly because the different split proportions of train, validation, and test sets on these datasets would lead to dramatically different comparison results (Dwivedi et al., 2020). We refer readers to (Dwivedi et al., 2020) for more details about the issues of these traditional datasets.

Table 4: Statistics of OGB LP datasets.

| Dataset | #Nodes | #Edges | #Degree |
|---|---|---|---|
| ogbl-ddi | $4,267$ | $1,334,889$ | 500 |
| ogbl-collab | $235,868$ | $1,285,465$ | 8 |
| ogbl-ppa | $576,289$ | $30,326,273$ | 73 |
| ogbl-citation2 | $2,927,963$ | $30,561,187$ | 21 |

As an alternative, we conduct experiments on four LP datasets from OGB (Hu et al., 2020). The statistics of datasets are summarized in Table 4.

**ogbl-ddi** is a homogeneous, unweighted, undirected graph built based on a drug-drug interaction network (Wishart et al., 2018). In this graph, a node represents a drug and an edge describes an interaction between two drugs. The nodes in this graph do not have any features. OGB officially splits edges into train, validations, and test sets according to what proteins those drugs target in the body. The test set is composed of drugs that predominantly bind to different proteins from drugs in the train and validation sets, thereby evaluating the generalization capacity of the model for practically useful LP.

**ogbl-collab** is an undirected graph extracted based on a collaboration network between authors from MAG (Wang et al., 2020). The nodes represent authors and the edges indicate the collaboration between authors. Every node has 128-dimensional features obtained by averaging the word embeddings of papers that are published by the authors (Hu et al., 2020). Each edge comes with an attribute, i.e., the *year* when the co-authored paper is published. The task is to predict future author collaborations. The data is split according to time, aiming to simulate a realistic application in collaboration recommendation.

**ogbl-ppa** is an undirected, unweighted graph. Nodes represent proteins from 58 different species, and edges are biologically meaningful associations between proteins (Szklarczyk et al., 2019). The node features are obtained based on the species. OGB splits the data according to the type of protein associations, which meets the practical needs.

**ogbl-citation2** is a directed graph built based on the citations between a subset of papers extracted from MAG (Wang et al., 2020). A node is a paper and has 128-dimensional features obtained by summarizing word embeddings of its title and abstract. A directed edge indicates that one paper cites another. The data is split according to time for the purpose of simulating a realistic application in citation recommendation.

