# OpenReview forum: "Can GNNs Learn Heuristic Information for Link Prediction?"
_ICLR.cc/2023/Conference — Submitted to ICLR 2023_

### Official Review · Reviewer_iMgJ · 2022-10-24

**Confidence:** 4
**Correctness:** 3
**Technical Novelty And Significance:** 1
**Empirical Novelty And Significance:** 2
**Recommendation:** 3

**Clarity, Quality, Novelty And Reproducibility:**

The presentation of the paper is easy to follow, but some parts are unclear, and the novelty is limited. See the above section on strengths and weaknesses.

**Strength And Weaknesses:**

Strengths

- The problems of GNN-based LP methods were analyzed through the comparison between the existing GNN-based LP methods and link heuristics.
- Experiments show superior performances of ComHG compared to existing GNN-based methods and link heuristics.

Weaknesses

- My main concern is the lack of novelty. Rather than proposing a GNN that can learn link heuristics, the authors simply embed the results of link heuristics and leverage them for link prediction by concatenating with representations produced by GNNs.

- Insufficient explanation of the proposed method. The author mentioned that embeddings of each heuristic method (SPD, CN, and RA) are fed into ComHG as inputs, but there is no specific explanation of how each method is embedded.

- The performance of GNN only in the ablation study (Fig 5.) and the performance of baselines using only GNNs (e.g., GCN, GraphSAGE, JKNet, GAT) are inconsistent. Specifically, the performances of only using GNN in ogbl-ppa and ogbl-ddi datasets are 61.24 and 92.15, respectively while performances of the baselines using only GNNs are under 24 and 58.


**Summary Of The Paper:**

This paper presents a simple yet light framework, ComHG, to predict the existence of a link by directly combining embeddings of link heuristics (e.g., Adamic Adar, Katz, RA) and the representations from a GNN. The authors further analyze that existing SEAL-type GNNs cannot effectively capture heuristic information. Experiments on four Open Graph Benchmark datasets show the effectiveness of ComHG for link prediction compared to existing GNN-based methods and link heuristics.

**Summary Of The Review:**

This paper presents a simple yet light framework for link prediction by directly combining embeddings of link heuristics and the representations from a GNN. The presentation of the paper is easy to follow, but the technical novelty is limited.

---

> ### Author Response · Authors · 2022-11-05
> **Response to reviewer iMgJ**
>
>
> Thanks for the comments.
>
> **1. My main concern is the lack of novelty. Rather than proposing a GNN that can learn link heuristics, the authors simply embed the results of link heuristics and leverage them for link prediction by concatenating with representations produced by GNNs.**
>
> First of all, we would like to highlight that our novel contributions are the theoretical analysis as shown in the Propositions of our paper. The proposed model is used to evaluate these propositions and is not the main contribution. We have emphasized this in the paper.
>
> We specifically designed ComHG using two separate modules instead of integrating GNNs and heuristics into one model, just aiming to verify our theoretical propositions. For example, in Proposition 1, GNNs and heuristics learn different information, in Proposition 3 SEAL cannot effectively combine GNN and heuristic information. These propositions motivated us to design the ComHG.
>
> **2. Insufficient explanation of the proposed method. The author mentioned that embeddings of each heuristic method (SPD, CN, and RA) are fed into ComHG as inputs, but there is no specific explanation of how each method is embedded.**
>
> Each heuristic method (SPD, CN, and RA) is a single numerical value and they can be encoded into latent embedding space using simple embedding techniques like word embeddings. Popular NN platforms like Pytorch, and TensorFlow have provided a simple function to do this. We follow the literature that rarely describes the embedding implementing process. We will add the embedding description in the appendix.
>
> **3. The performance of GNN only in the ablation study (Fig 5.) and the performance of baselines using only GNNs (e.g., GCN, GraphSAGE, JKNet, GAT) are inconsistent. Specifically, the performances of only using GNN in ogbl-ppa and ogbl-ddi datasets are 61.24 and 92.15, respectively while performances of the baselines using only GNNs are under 24 and 58.**
>
> Firstly, we follow the results of these baselines in the official leaderboard https://ogb.stanford.edu/docs/leader_linkprop/. Secondly, our model performs better because we use trainable node embeddings. The link predictor setting is also different from the baselines. We will report the results of our model only using GNN to the OGB official team to update their baseline results.

---

> > ### Public Comment · ~Juanhui_Li1 · 2022-11-07
> > **Question of the incomplete code**
> >
> > Dear authors, we are quite interested in your paper according to its surprising good performance on the ogb datasets (especially on the ogbl-ddi and ogbl-ppa). We checked the code link given in the paper but found the code is incomplete. Could you provide the the complete code? It would be very helpful for our current work. Any help is appreciated!

---

> > > ### Author Response · Authors · 2022-11-07
> > > **Code availabe**
> > >
> > > Hi Juanhui,
> > >
> > > Thanks for your appreciation. We will open all code soon (maybe after the stage of double-blind reviewing).

---

> > > > ### Public Comment · ~Juanhui_Li1 · 2022-11-08
> > > > **Question of encoding RA**
> > > >
> > > > Dear authors, we have a question for embedding the RA scores. From your second response to the reviewer iMgj, we guess you use the look up operation to get the embedding for SPD and CN? It makes sense for SPD and CN because they are the discrete values. But how to implement for RA considering it's continuous? Thanks!

---

> > > > > ### Author Response · Authors · 2022-11-08
> > > > > **encoding RA, AA ...**
> > > > >
> > > > > Hi Juanhui,
> > > > >
> > > > > RAs, AAs, etc., are continuous values and most of them are located between a specific interval. We can separate this interval into small bins and give an embedding vector to each bin. Besides, we only encode each bin when the value is small. For the heuristics, including RA, AA, SPDs, or CNs that are larger than a given value, we encode them into the same embedding vector. We investigate the distribution of the values of each heuristic and then based on the distribution, set the bins or the given value. Our experiments also confirm such simple encoding methods are efficient.

---

### Official Review · Reviewer_gAm5 · 2022-10-25

**Confidence:** 5
**Clarity, Quality, Novelty And Reproducibility:** The paper is clear. The novelty is mo…
**Correctness:** 3
**Technical Novelty And Significance:** 3
**Empirical Novelty And Significance:** 3
**Recommendation:** 6

**Strength And Weaknesses:**

Pros: The authors intuitively show that traditional link heuristics and GNNs capture different information. They analyze this finding in theory and conduct extensive experiments to distinguish the superior performance of their proposed method.

Cons: Authors mainly analyze the subgraph based graph neural networks and point out their limitations. There are also node-wise embedding based graph neural networks which are more efficient and effective than the subgraph based ones. The authors are suggested to discuss them in the motivation. Moreover, efficiency is another important metric for evaluation link prediction systems. The authors are suggested to evaluate the inference efficiency of their method.

**Summary Of The Paper:**

The authors question whether these methods can effectively learn the information equivalent to link heuristics such as Common Neighbors, Katz index, etc. (we refer to such information as heuristic information in this work). The authors show that link heuristics and GNNs capture different information. Link heuristics usually collect pair-specific information by counting the involved neighbors or paths between two nodes in a candidate link, while GNNs learn node-wise representations through a neighborhood aggregation algorithm in which two nodes
in the candidate link do not pay special attention to each other.

**Summary Of The Review:**

Please refer to the above points for improvements.

---

> ### Author Response · Authors · 2022-11-05
> **Response to reviewer gAm5**
>
>
> Thanks for the comments.
>
> **1. There are also node-wise embedding-based graph neural networks which are more efficient and effective than the subgraph based ones.**
>
> We guess the node-wise embedding-based methods in this question are GCN, GAT, etc. Actually, Proposition 2 has analyzed such methods.
>
> **2. Efficiency is another important metric for evaluation link prediction systems.**
>
> We agree with the reviewer. Efficiency is important in link prediction training. We have theoretically analyzed the computational complexity in the last paragraph in Section 4.1. The complexity of the proposed ComHG is on par with the traditional GNNs like GCN, and is much lighter than subgraph-based methods like SEAL.

---

### Official Review · Reviewer_ugPY · 2022-10-29

**Confidence:** 5
**Correctness:** 3
**Technical Novelty And Significance:** 2
**Empirical Novelty And Significance:** 2
**Recommendation:** 5

**Clarity, Quality, Novelty And Reproducibility:**

The paper is well-written and easy to follow.
Not sure if the readers can reproduce the proposed model by following the paper.
Is the code of the proposed model publicly available?

**Strength And Weaknesses:**

Strengths of the paper:
Well-written and easy to follow.
Experiments are conducted on a number of datasets.

Weaknesses and main concerns of the paper:

The authors claim that “the information captured by link heuristics is different from that embedded in the node representations learned by GNNs.”. What’s the relationship between the information by link heuristics and GNNs then? Are they independent to each other?

It is unclear why the authors only focus on link prediction but not other tasks such as node classification, as the inferred embeddings can be utilized in other tasks.

It is unclear if the link heuristics can be simply defined as in Equation (1).

The contribution of the proposed model, ComHG, is marginal, which is straightforward and built based on traditional GNN model and a traditional Encoder. In terms of GNN model and the Encoder models, the contribution is marginal.

Other baselines can be taken into account for comparisons such as:
Bouritsas et al., Improving graph neural networks expressivity via subgraph isomorphism counting, TPAMI, 2022.
Fang et al., Structure-aware random fourier kernel for graphs, in NeurIPS 2021.

Are there other information that are different from “link heuristic” and “node-wise attributes” that can be integrated into ComHG?

Other minor concerns:
The first two paragraphs in the introduction section can be deleted, as most machine learning researchers have the background knowledge. (up to you)

What’re the motivations to investigate the problem: whether the GNN-based LP methods can effectively learn the information equivalent to link heuristics for LP? Can the answers and analysis inspire research on other topics?

Can we use graph neural network distillation techniques to address the task?


**Summary Of The Paper:**

The authors propose a light link prediction framework ComHG by combining link heuristics and the GNN. Link heuristics in CombHG are encoded into trainable embeddings and combined with the representations produced by a GNN.

**Summary Of The Review:**

The authors are encouraged to address the issues raised in the above.

---

> ### Author Response · Authors · 2022-11-05
> **Response to reviewer ugPY**
>
>
> Thanks for the comments.
>
> **1. What’s the relationship between the information by link heuristics and GNNs then? Are they independent to each other?**
>
> We say GNNs can hardly count the number of shared neighbors or paths between two nodes but link heuristics can. We are concerned more about the difference between GNNs and heuristics instead of the similarity between them. This is because integrating different information rather than similar information can improve the link prediction performance.
>
> **2. It is unclear why the authors only focus on link prediction but not other tasks such as node classification**
>
> Because node classification does not involve the link heuristic information related to the common neighbors or paths between two nodes.
>
> **3. It is unclear if the link heuristics can be simply defined as in Equation (1)**
>
> We have presented two heuristic examples that can be generalized by Eq 1.
>
> **4. The contribution of the proposed model, ComHG, is marginal, which is straightforward**
>
> Our ComHG is specifically designed to evaluate our theoretical propositions which are our main contributions. In our paper, we have highlighted that our model is straightforward.
>
> **5. Other baselines can be taken into account for comparisons**
>
> We mainly outline the baselines verified in the OGB official leaderboard https://ogb.stanford.edu/docs/leader_linkprop/. These baselines have covered the mainstream LP methods.
>
> **6. Can we use graph neural network distillation techniques to address the task?**
>
> No, because as discussed in Proposition 1, GNNs lack the model ability to calculate the number of involved nodes or paths between two nodes in a candidate link node pair.
>
> **7. Is there other information that is different from “link heuristic” and “node-wise attributes” that can be integrated into ComHG?**
>
> This is an interesting research direction. However, this paper focus on the ability of GNN to learn heuristic information. We present ComHG with the main purpose of evaluating our propositions on this topic. Adding other types of information would bias the experimental evaluation of our theoretical propositions.
>
> **8.Other minor concerns: The first two paragraphs in the introduction section can be deleted, as most machine learning researchers have the background knowledge. (up to you)**
>
> Thanks for your suggestion. We will revise it.
>
> **9. What’re the motivations to investigate the problem: whether the GNN-based LP methods can effectively learn the information equivalent to link heuristics for LP? Can the answers and analysis inspire research on other topics?**
>
> Intuitively, GNNs can hardly capture the information related to the number of common neighbors or paths between a node pair but link heuristics can. This is our motivation. Figure 1 in the introduction also illustrates this. Our following propositions provide in-depth analyses of this topic.
>
> Our paper shows GNNs and link Heuristics capture different information theoretically and empirically. Considering the promising performance of our model, this work can inspire research in multiple-node tasks like link prediction, recommendation systems in knowledge graphs, and even graph-level classification.

---

### Official Review · Reviewer_HJFA · 2022-10-30

**Confidence:** 4
**Correctness:** 3
**Technical Novelty And Significance:** 2
**Empirical Novelty And Significance:** 2
**Recommendation:** 5

**Clarity, Quality, Novelty And Reproducibility:**

The paper is clearly written. I may have missed the code which I was hoping to look into. Was that made available?
In terms of the technical quality, I would say there are scopes of improvements--- see below.  The paper lacks significant novelty, I think it is mainly because of the richness of this area.

**Strength And Weaknesses:**

Strengths:
+ Good literature survey
+ Experiments are performed with due diligence

Weaknesses:
- Several issues with experiments (see the detailed review)
- Some important baselines (https://arxiv.org/abs/2110.04375, https://arxiv.org/abs/2012.08974) are missing
- No code

**Summary Of The Paper:**

The authors proposes if GNNs (which typically take into account neighborhood information and aggregates then, can also utilize heuristic measures like pairwise node information for link Prediction tasks and whether that improves vanilla GNN for link Prediction. The author states through experiments that a combination of link heuristics and GNN-based neighborhood aggregation techniques results in comparatively better link Prediction than only vanilla GNN and heuristic-based models.


**Summary Of The Review:**

I elaborate the strengths and weaknesses here:

Strengths
+ The background setting is established well for the importance of Link Prediction and its wide applications.  The authors have performed experiments on large datasets (typically Order of millions in nodes) and they have showcased empirically the comparative performance of their proposed methods w.r.t GNN based Link heuristic methods (like SOTA - SEAL), GNN based methods (SOTA - GraphSage).
+ The ablation study at the end of the effect of link heuristics and GNN modules in the proposed method is well investigated and clearly shows how sparsity of the graph effects each of the components contribution. The authors provide a strong reasoning behind this phenomenon and its empirically showcased.
+ Model complexity wise the author theoretically shows how their proposed method has less complexity than SOTA SEAL method.


Weakness:

-  Missing important baselines https://arxiv.org/abs/2110.04375, https://arxiv.org/abs/2012.08974

- The question raised in the introduction part is valid but the authors do not clearly convey what methods like SEAL fail to capture which traditional link based heuristics capture. That information could setup a more strong motivational setting for the rest of the paper to follow . Why is it important to capture link based heuristics ? Does it show significant improvement than GNNs which fail to capture such info. If so any empirical reference or short writing as to why it is so.

- The author mentions that through experimental results they are able to show the gap why link based heuristics are important for LP. It would have been nice to showcase the gap also theoretically. In link heuristics only a few classical methods are mentioned while other important link based heuristics like simrank / simrank* and it's variations are not mentioned.

- The generalization into the set structure is a bit unconvincing like in typical simrank computations which takes into account recursiveness of its computation for all its neighbours. The fixed one hop is only assumed. Also the definition for setop nullifies there.

- Proposition I. states that GNNs doent take into account the attenetion for a particular node or pairwise. However, models like GAT have been proposed to take that into account. What if GNNs that take into account edge reperesnetations like NBFNet ( although here it's mentioned for only node level representationsl) then do we require such heuristics any more ? What if we had hybrid of NBFNet edge emebeddings along with traditional node representation based GNNs. Does that solve the problem? Did the authors perform experiments on those?

- Proposition I  is not convincing.  Do the authors share any examples where the pooling information over elements in one set don't contribute? A pooling information over a set of neighborhoods of a particular node will also have the inherent pooling information of the individual neighborhood nodes via message passing. Does that information. Not similar in essence to link Prediction tasks. If not can it theoretically be shown or via suitable examples.

- Proposition II also lacks the same mathematical rigour and explanation as to why combine operation doesn't capture the same info as compared to a set intersection.

- Proposition III is well explained via an example. However we can use weighted attentetion or weighted mean in the pooling strategy or other pooling strategies can also be used to countereffect the loss in information. Only SEAL GNN based LP was analysed. But not any other GNN based LP. For the propositions to hold it needs to be shown for all GNN based LPs and should be mathematically motivated as to which information gets missed if we have a clever pooling strategy. Also empirical results should be cited in those propositions stating whether SEAL based LP performs poorly for certain examples.

- Does combHGs heuristic embedding not lose information since it basically a concatenation of node embeddings and as per previous propositions it should lose some information. The node embeddings might change later on due to the GNN pooling. It's not clear.

- As per table 2 if GNN based LP produces better than traditional LPs then it contradicts the proposition that GN based LP fail to capture those heuristics which is a fundamental notion of this paper ( why such heuristics are important).

- I think Hits are not so crucial metric in the context of LP, it does not capture the recall. MAP is better.

---

> ### Author Response · Authors · 2022-11-05
> **Response to reviewer HJFA**
>
> Thanks for the comments.
>
> **1. Missing important baselines.**
>
> We mainly outline the baselines verified in the OGB official leaderboard https://ogb.stanford.edu/docs/leader_linkprop/. These baselines have covered the mainstream LP methods.
>
> **2. The question raised in the introduction is valid but is not explained clearly.**
>
> We leave detailed explanations in the theoretical and experimental sections. It is a good suggestion to add a further explanation for the questions in the introduction.
>
> **3. Other important link based heuristics like simrank are not mentioned. The generalization into the set structure is a bit unconvincing like in simrank computations.**
>
> We agree with the reviewer. The generalization in our Eq 1 can generalize many heuristics based on the neighborhood (CN, AA) or path (SPD, Katz) but not random-walk-based heuristics like SimRank, PageRank, etc. Generalizing all types of heuristics is not the main purpose of our paper. Instead, we focus on why GNN cannot learn information equivalent to several popular link heuristics.
>
> **4. Proposition I. states that GNNs don't take into account the attention of a particular node or pairwise. However, models like GAT can.**
>
> Our main claim in proposition 1 is that GNNs can hardly calculate the number of shared neighbors or paths (i.e. heuristics) between two nodes in a candidate link. In this regard, GAT also fails to do such counting because GAT still follows the neighborhood aggregation algorithm.
>
> **5. What if GNNs that take into account edge representations like NBFNet then do we require such heuristics? What if we had hybrid of NBFNet edge embeddings along with traditional node representation? Does that solve the problem? Did the authors perform experiments on those?**
>
> We also have the same concern. NBFNet is a special network that approximates several path-based heuristics while having less ability to capture neighborhood-based heuristics like CN, AA. Combining NBFNet with node representations is an interesting research direction, especially considering the promising performance of our simple ComHG. However, this paper aims to prove that the GNNs cannot learn the information equivalent to link heuristics for LP. Extending NBFNet is out of our main purpose. Besides, we present our simple ComHG mainly because of verifying our theoretic propositions, where ComHG uses two separate modules, i.e GNN and heuristic module, to investigate the difference between GNNs and heuristics.
>
> **6. Proposition I is not convincing. Do the authors share any examples where the pooling information over elements in one set doesn't contribute?**
>
> Intuitively, GNNs can hardly capture the information related to the number of common neighbors or paths between a node pair. However, we cannot say that the representations learned by GNNs (i.e. pooling information over elements) do not contain any heuristic information. It is difficult to interpret the representations learned by GNNs.
>
> **7. Proposition II also lacks the same mathematical explanation as to why the combine operation doesn't capture the same info as compared to a set intersection.**
>
> Simply combining two nodes' representations cannot learn the information captured by set intersections like common neighbors because such a combination does not involve counting the shared neighbors or paths between two nodes. We will add a more concrete explanation to illustrate this in propositions 1 and 2.
>
> **8. We can use weighted attention or weighted mean in the pooling strategy in SEAL**
>
> Attention-based pooling only assigns different weights in the GNN aggregation but can hardly calculate the number of the shared neighbors or paths between two nodes which is the core operation in heuristics. Therefore, weighted mean pooling in SEAL still suffers from the same issue as non-attention-based pooling.
>
> **9. Empirical results should be cited in those propositions**
>
> We will add citations of experimental evidence in those propositions.
>
> **10. The node embeddings might change the heuristic embeddings due to the GNN pooling. It's not clear.**
>
> It is clear. The heuristics embeddings are not involved in the GNN module and can not be 'changed' by the GNN pooling. The heuristics embeddings and GNNs are trained using two separate modules.
>
> **11. If GNN-based LP produces better than traditional LPs then it contradicts the proposition that GN-based LP fails to capture those heuristics.**
>
> We disagree with this comment. The better performance of GNNs only confirms that GNNs learn more effective information but cannot confirms that GNNs have learned heuristic information. In our paper, we prove that GNNs and heuristics capture different but effective information for LP.
>
> **12. I think Hits are not so crucial metric in the context of LP, it does not capture the recall. MAP is better.**
>
> We follow the standard evaluation procedure in the official OGB. Please check https://ogb.stanford.edu/docs/leader_linkprop/

---

### Decision · Program_Chairs · 2023-01-20

**Decision:**

Reject

**Justification For Why Not Higher Score:**

The proposed method is straightforward and lack a detailed analysis what kind of information by link heuristics GNN cannot learn. Experiments and analysis are all on macro-level.

**Justification For Why Not Lower Score:**

N/A

**Metareview: Summary, Strengths And Weaknesses:**

Heuristic and SEAL-like methods are popular ones for link prediction. Authors theoretically show that SEAL-like methods cannot effectively capture heuristic information, which can be a potential problem to get better performance. Based on this observation, the authors propose to concat embeddings from heuristic and SEAL-like methods. Experiments on four OGB datasets have shown significant improvements can be achieved compared with existing methods.

Strength
- Theoretical analysis is provided.
- Literature review is extensive and well-organized.

Weakness
- The proposed method is simple.
- Insufficient analysis on the motivation of the paper.